# BOOSTING ADVERSARIAL TRAINING WITH MASKED ADAPTIVE ENSEMBLE

## ABSTRACT

Adversarial training (AT) can help improve the robustness of a deep neural network (DNN) against potential adversarial attacks by intentionally injecting adversarial examples into the training data, but this way inevitably incurs standard accuracy degradation to some extent, thereby calling for a trade-off between standard accuracy and robustness. Besides, the prominent AT solutions are vulnerable to sparse attacks, due to "robustness overfitting" upon dense attacks, often adopted by AT to produce a threat model. To tackle such shortcomings, this paper proposes a novel framework, including a detector and a classifier bridged by our newly developed adaptive ensemble. Specifically, a Guided Backpropagation-based detector is designed to sniff adversarial examples, driven by our empirical observation. Meanwhile, a classifier with two encoders is employed for extracting visual representations respectively from clean images and adversarial examples. The adaptive ensemble approach also enables us to mask off a random subset of image patches within input data, eliminating potential adversarial effects when encountering malicious inputs with negligible standard accuracy degradation. As such, our approach enjoys improved robustness, able to withstand both dense and sparse attacks, while maintaining high standard accuracy. Experimental results exhibit that our detector and classifier outperform their state-of-the-art counterparts, in terms of detection accuracy, standard accuracy, and adversarial robustness. For example, on CIFAR-10, our detector achieves the best detection accuracy of $99.6\%$ under dense attacks and of $98.5\%$ under sparse attacks. Our classifier achieves the best standard accuracy of $91.2\%$ and the best robustness against dense attack (or sparse attack) of $57.5\%$ (or $54.8\%$).

## 1 INTRODUCTION

Deep neural networks (DNNs) have been reported to be vulnerable to adversarial attacks. That is, maliciously crafting clean images under a small distance can mislead DNNs into incorrect predictions. Such vulnerability prevents DNNs' wide adoption in critical domains, such as healthcare, autonomous driving, finances, among many others. In a nutshell, adversarial attacks can be roughly grouped into two categories, *i.e.*, the dense attack and the sparse attack. The former (*e.g.*, Goodfellow et al. (2015); Moosavi-Dezfooli et al. (2016); Madry et al. (2018); Croce & Hein (2020); Yao et al. (2021)) tends to perturb almost all pixels on the clean image, whereas the latter (*e.g.*, Papernot et al. (2016); Carlini & Wagner (2017); Modas et al. (2019); Dong et al. (2020); Pintor et al. (2021); Zhu et al. (2021)) modifies only a limited number of pixels to fool the DNN models.

So far, adversarial training (AT) is widely accepted as the most effective method to improve DNNs' robustness against adversarial attacks, by intentionally injecting adversarial examples into the training data. In particular, multi-step ATs Madry et al. (2018); Zhang et al. (2019); Jia et al. (2022) perform multi-step dense attacks (*e.g.*, PGD attack) to find the worst-case adversarial examples for training, achieving state-of-the-art robustness but incurring a significant computational overhead. On the other hand, by using the single-step dense attack (*e.g.*, FGSM attack), one-step ATs Wong et al. (2020); Andriushchenko & Flammarion (2020); Kim et al. (2021); Li et al. (2022); Wang et al. (2022) can significantly reduce the computational overhead while achieving decent robustness under dense attacks. Despite effectiveness, existing ATs suffer from two shortcomings: i) a trade-off between standard accuracy (*i.e.*, the accuracy on clean images) and adversarial robustness (*i.e.*, the accuracy on adversarial examples), with improved robustness yielding non-negligible standard

accuracy degradation and ii) robustness overfitting on dense attacks, making improved robustness vulnerable to sparse attacks.

One promising direction to address the trade-off between standard accuracy and adversarial robustness is via a *detection/rejection* mechanism, that is, training an additional detector to reject malicious input data, with various detection techniques proposed Roth et al. (2019); Ma & Liu (2019); Yin et al. (2020); Raghuram et al. (2021); Tramèr (2022). Unfortunately, the *detection/rejection* mechanism is still ineffective to defend against sparse attacks, as sparse attacks only perturb a limited number of pixels. Even worse, the *detection/rejection* mechanism can be applied merely to a limited number of scenarios. For example, it cannot be generalized to the application domains where natural adversarial examples exist, as reported in a recent study Hendrycks et al. (2021).

In this work, we consider the robustness under a more general and challenging scenario (than that addressed earlier by the detection/rejection mechanism), where the malicious input is not allowed to be rejected. Note that a robust model under such a scenario is crucial for applying DNNs to critical domains. For example, an autonomous driving car is expected to recognize a road sign even if it has been maliciously crafted. Our goal is to develop a novel framework, including a detector and a classifier, to boost adversarial training for improving DNN's robustness against both dense and sparse attacks at a small expense of standard accuracy degradation. Specifically, our framework is adversarially trained by using one-step least-likely adversarial training, adopted from Fast Adversarial Training Wong et al. (2020) with slight modification (see Section A.2 in Appendix for details). We incorporate two new designs in our detector to make adversarial examples more noticeable. First, we resort to Guided Backpropagation Springenberg et al. (2015) to expose adversarial perturbations, driven by our empirical observations. Second, the Soft-Nearest Neighbors Loss (SNN Loss) Salakhutdinov & Hinton (2007); Frosst et al. (2019) is tailored to push adversarial examples away from their corresponding clean images. As such, our detector is effective in sniffing both dense attack-generated and sparse attack-generated adversarial examples.

Our classifier includes two encoders for extracting visual representations respectively from clean images and adversarial examples, aiming to alleviate the negative effect of adversarial training on standard accuracy. We separate the training process into "pre-training" and "fine-tuning" for representation learning and classification, respectively. In the pre-training, our goal is to jointly learn high-quality representations and encourage pairwise similarity between a clean image and its adversarial example. Specifically, we extend Masked Autoencoders (MAE) He et al. (2022), *i.e.*, learning visual representations by reconstructing the masked images, for adversarial training via a new design. That is, we reconstruct images from a pair of masked clean image and masked adversarial example, for representation learning, with a contrastive loss on visual representations to encourage pair similarity. In the fine-tuning of classification, we freeze the weights on the two encoders and fine-tune an MLP (Multi-layer Perceptron) for accurate classification by using our proposed adaptive ensemble to bridge the detector and the classifier. Meanwhile, our adaptive ensemble allows us to mask off an arbitrary subset of image patches within the input, enabling our approach to mitigate potential adversarial effects when encountering malicious inputs with negligible standard accuracy degradation. Extensive experiments have been carried out on three popular benchmarks, with the results demonstrating that our solutions outperform state-of-the-art detection and adversarial training techniques in terms of detection accuracy, standard accuracy, and robustness.

## 2 RELATED WORK

Our work closely relates to two research scopes, *i.e.*, detection/rejection mechanisms and adversarial training approaches. This section discusses how our work relates to, and differs from, prior studies.

**Detection Mechanisms.** Detecting adversarial examples (AEs) and then rejecting them (*i.e.*, detection/rejection mechanism) can improve the model robustness. That is, the input will be rejected if the detector classifies it as an adversarial example. Popular detection techniques include Odds Roth et al. (2019), which considers the difference between clean images and AEs in terms of log-odds; NIC Ma & Liu (2019), which checks channel invariants within DNNs; GAT Yin et al. (2020), which resorts to multiple binary classifiers; JTLA Raghuram et al. (2021), which proposes a detection framework by employing internal layer representations, among many others Lee et al. (2018); Yang et al. (2020); Sheikholeslami et al. (2021). Unfortunately, existing detection methods are typically ineffective in sniffing sparse attack-generated AEs, which just modify limited numbers of pixels.

Besides, the detection/rejection mechanism only works in limited scenarios. For example, it cannot be generalized to domains where natural adversarial examples exist. Differently, our work resorts to the Guided-Backpropogation technique, which can largely expose adversarial perturbations, based on our empirical observation. Then, we adopt the Soft-Nearest Neighbors (SNN) loss, which can further maximize differences between clean images and adversarial examples.

**Adversarial Training Approaches.** Adversarial training (AT) aims to improve the model robustness by intentionally injecting adversarial examples into the training data. For example, PGD-AT Madry et al. (2018) proposes a multi-step attack to find the worst case of training data, TRADES Zhang et al. (2019) addresses the limitation of PGD-AT by utilizing theoretically sound classification-calibrated loss, EAT Tramèr et al. (2018) uses an ensemble of different DNNs to produce the threat model, FAT Wong et al. (2020) reduces the computational overhead of AT by utilizing FGSM attack with the random initialization, LAS-AWP Jia et al. (2022) boosts AT with a learnable attack strategy, Sub-AT Li et al. (2022) constrains AT in a well-designed subspace, and many others Shafahi et al. (2019); Andriushchenko & Flammarion (2020); Kim et al. (2021); Wang et al. (2022). However, prior ATs suffer from the dilemma of balancing the trade-off between standard accuracy and adversarial robustness. Besides, their improved robustness is vulnerable to sparse attacks. Although we adopt the threat model in FAT, by using the proposed adaptive ensemble, our method can be generalized to defend against sparse attacks. Meanwhile, it mitigates the standard accuracy degradation by employing two encoders for extracting visual representations respectively from clean images and adversarial examples.

# 3 OUR APPROACH

## 3.1 PROBLEM STATEMENT

We consider a set of $N$ samples, $i.e.$, $\mathbb{X} = \{(\boldsymbol{x}_i, \ y_i) \mid i \in \{1, 2, \ldots, N\}\}$, where $\boldsymbol{x} \in \mathbb{R}^{H \times W \times C}$ is the input image and $y \in [C]$ denotes its label. Here, $(H, W)$ represents the resolution of input images and $C$ is the number of channels. For notational convenience, we let $d = H \times W \times C$. A classifier is a function $f_{\boldsymbol{\theta}}: \mathbb{R}^d \to [C]$, parameterized by a neural network. In this paper, we consider two types of inputs, $i.e.$, the clean image $\boldsymbol{x}^{\text{cln}}$ sampled from the standard distribution $\mathcal{D}_{\text{std}}$ and the adversarial example $\boldsymbol{x}^{\text{adv}}$ sampled from the adversarial distribution $\mathcal{D}_{\text{adv}}$. We assume $\mathcal{D}_{\text{std}}$ and $\mathcal{D}_{\text{adv}}$ follow different distributions. The clean image $\boldsymbol{x}^{\text{cln}}$ itself or its augmented variant can be the input, while the adversarial example $\boldsymbol{x}^{\text{adv}}$ is a malicious version of $\boldsymbol{x}$ within a small distance, that is, for some metric $d$, we have $d(\boldsymbol{x}, \boldsymbol{x}^{\text{adv}}) \leq \epsilon$, but $\boldsymbol{x}^{\text{adv}}$ can mislead conventional classifiers. Parameterized by another neural network, a detector $g_{\boldsymbol{\phi}}$ is to tell whether an input image is a clean image or not, $i.e.$, $g_{\boldsymbol{\phi}}: \mathbb{R}^d \to \{\pm 1\}$, where $+1$ and $-1$ indicate the clean image and the adversarial example, respectively. The binary indicator function $\mathbb{1}_{\{\cdot\}}$ is 1 if both the detector $g_{\boldsymbol{\phi}}$ and the classifier $f_{\boldsymbol{\theta}}$ make correct predictions. In this paper, we follow previous studies Madry et al. (2018); Zhang et al. (2019) by referring standard accuracy $\mathcal{A}_{\text{std}}$ and adversarial robustness $\mathcal{A}_{\text{adv}}$, as classification accuracy on clean images and on adversarial examples, respectively.

We start by defining standard accuracy. Given a clean image, the incorrect prediction made either by the detector or by the classifier is counted as an error.

**Definition 3.1** (Standard accuracy). Let $f \circ g$ be a model with a classifier $f_{\boldsymbol{\theta}}: \mathbb{R}^d \to [C]$ and a detector $g_{\boldsymbol{\phi}}: \mathbb{R}^d \to \{\pm 1\}$. Its standard accuracy is defined by the expected rate at which both the detector and the classifier make correct predictions on the clean image:

$$\mathcal{A}_{\text{std}}(f \circ g) := \mathop{\mathbb{E}}_{(\boldsymbol{x}^{\text{cln}}, y) \sim \mathcal{D}_{\text{std}}} \left[ \mathbb{1}_{\{g_{\boldsymbol{\phi}}(\boldsymbol{x}^{\text{cln}})=1 \ \wedge \ f_{\boldsymbol{\theta}}(\boldsymbol{x}^{\text{cln}})=y\}} \right]. \tag{1}$$

Similar to standard accuracy, adversarial robustness regards the incorrect prediction on an adversarial example made either by the detector or by the classifier as an error.

**Definition 3.2** (Adversarial robustness). Let $f \circ g$ be a model, including a detector $g_{\boldsymbol{\phi}}: \mathbb{R}^d \to \{\pm 1\}$ and a classifier $f_{\boldsymbol{\theta}}: \mathbb{R}^d \to [C]$. Given the input $(\boldsymbol{x}, y)$, the adversarial robustness at a small distance $\epsilon$, $i.e.$, $d(\boldsymbol{x}, \boldsymbol{x}^{\text{adv}}) \leq \epsilon$, is defined as:

$$\mathcal{A}_{\text{adv}}^{\epsilon}(f \circ g) := \mathop{\mathbb{E}}_{(\boldsymbol{x}^{\text{adv}}, y) \sim \mathcal{D}^{\text{adv}}} \left[ \mathbb{1}_{\{g_{\boldsymbol{\phi}}(\boldsymbol{x}^{\text{adv}})=-1 \ \wedge \ f_{\boldsymbol{\theta}}(\boldsymbol{x}^{\text{adv}})=y\}} \right]. \tag{2}$$

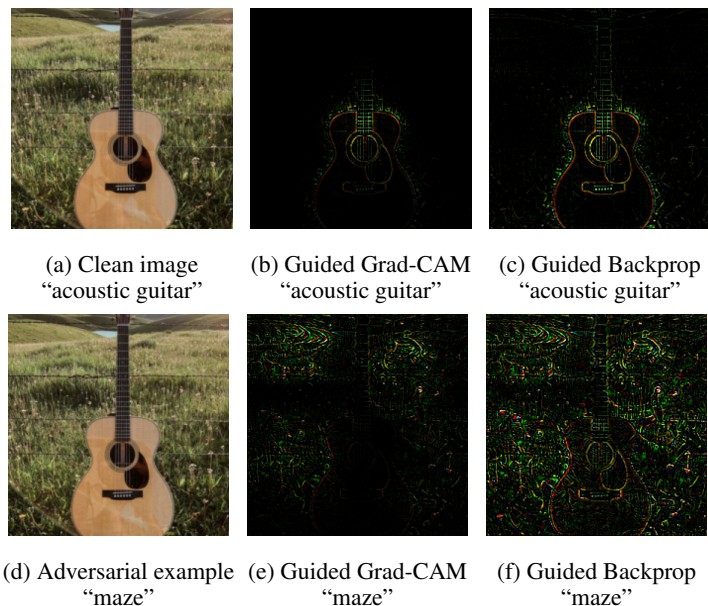

(a) Clean image
"acoustic guitar"

(b) Guided Grad-CAM
"acoustic guitar"

(c) Guided Backprop
"acoustic guitar"

(d) Adversarial example
"maze"

(e) Guided Grad-CAM
"maze"

(f) Guided Backprop
"maze"

Figure 1: Different visualizations on the clean image (Top) and the adversarial example (Bottom). **Left:** Original clean image and adversarial example with their predicted labels. **Middle:** Guided Grad-CAM visualization. **Right:** Guided Backpropagation visualization.

### 3.2 DETECTION

Parameterized by a neural network with parameters $\phi$, the detector $g_\phi : \mathbb{R}^d \to \{\pm 1\}$ is to determine whether the input is a clean image or not, where $+1$ and $-1$ indicate the clean image and the adversarial example, respectively. Mathematically,

$$g_\phi(\boldsymbol{x}) = \begin{cases} +1, & \text{if } \boldsymbol{x} \text{ is a clean image} \\ -1, & \text{otherwise.} \end{cases} \tag{3}$$

Aiming to generalize the robust model to critical domains (*e.g.*, autonomous driving), the input will not be rejected in this study. That is, the detector also outputs the estimated probability of $p \in [0, 1]$ for the clean image and $1 - p$ for the adversarial example [1].

The design of our detector architecture is motivated by our empirical observation in that *the adversarial perturbation is detectable after Guided Backpropagation visualization.* Due to the small distance between a clean image and its corresponding adversarial example, their difference is notoriously imperceptible (see Figures 1a and 1d), making it theoretically hard to detect adversarial examples Tramèr (2022). In our empirical study, we resort to Guided Backpropagation Springenberg et al. (2015) to visualize the difference between the clean image and the adversarial example. Surprisingly, we discovered that after Guided Backpropagation visualization on the adversarial example, its adversarial perturbation is quite noticeable; See Figure 1c versus Figure 1f, *i.e.*, visualization on the clean image versus on the adversarial example. Notably, our experiments also include the visualization comparison under Guided Grad-CAM Selvaraju et al. (2017), developed recently; see Figure 1b versus Figure 1e. However, Guided Grad-CAM exhibits inferior performance (compared to Guided Backpropagation) in terms of exposing adversarial perturbation. This empirical study motivates us to maximize the difference between clean images and adversarial examples by using Guided Backpropagation visualization.

Figure 2 illustrates our detector architecture. Given an input image $\boldsymbol{x} \in \mathbb{R}^d$, we perform Guided Backpropagation on the original image, arriving at an input variant $\boldsymbol{x}' \in \mathbb{R}^d$. Note that we employ the label predicted by a pre-trained model to be the target concept for Guided Backpropagation; hence, no ground-truth label is required during the detection. Following the standard Vision Transformer (ViT) Dosovitskiy et al. (2021), we patchify the two inputs into two sets of image patches

---

[1]Notably, our approach is readily generalizable to the detection/rejection mechanism. We can reject the input data if the detector identifies it as a malicious input, *i.e.*, $g_\phi(\boldsymbol{x}) = -1$.

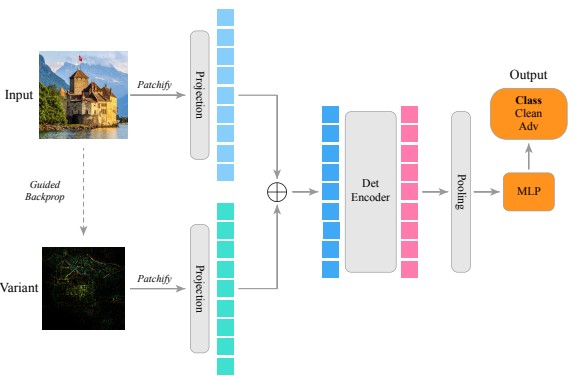

Figure 2: Our detector architecture.

and embed them via linear projection, arriving at two sets of patch embeddings, *i.e.*, $\mathbf{E}_p \in \mathbb{R}^{M \times D}$ and $\mathbf{E}'_p \in \mathbb{R}^{M \times D}$, respectively for the original input and its input variant. Here, $M$ represents the number of patches and $D$ indicates the hidden dimension. Driven by the above empirical observation, we add two sets of patch embeddings together, expecting that the adversarial perturbation exposed by Guided Backpropagation can help differentiate adversarial examples from clean images. Similar to the standard ViT, we fill the position embeddings $\mathbf{E}_{\text{pos}} \in \mathbb{R}^{M \times D}$ for remaining positional information. Hence, we have the input sequence $\mathbf{E}$ of the detection encoder (*i.e.*, a Transformer Encoder) as follows,

$$\mathbf{E} = [\mathbf{E}_{p_1} + \mathbf{E}'_{p_1};\ \mathbf{E}_{p_2} + \mathbf{E}'_{p_2}; \cdots;\ \mathbf{E}_{p_M} + \mathbf{E}'_{p_M}] + \mathbf{E}_{\text{pos}}, \tag{4}$$

where $\mathbf{E}_{p_i}$ (or $\mathbf{E}'_{p_i}$) denotes the $i$-th patch embedding in the original input (or its input variant produced by Guided Backpropagation). Following the ViT architecture in Masked Autoencoders (MAE) He et al. (2022), we perform the global average pooling on the full set of encoded patch embeddings, with the result fed into an MLP (*i.e.*, multiple-layer perceptron) for telling whether the input is a clean image or not.

Aiming to further differentiate adversarial examples from clean images, we propose a novel loss function to train our detector, including a Cross-Entropy (CE) Loss $\mathcal{L}_{\text{ce}}$ and a Soft-Nearest Neighbors (SNN) loss $\mathcal{L}_{\text{snn}}$ Salakhutdinov & Hinton (2007); Frosst et al. (2019), for jointly penalizing the detection error and the similarity level between the clean image and the adversarial example, *i.e.*,

$$\mathcal{L}_{\text{det}} = (1 - \lambda) \cdot \mathcal{L}_{\text{ce}}(g_\phi(\boldsymbol{x}), y^{\text{det}}) + \lambda \cdot \mathcal{L}_{\text{snn}}(\boldsymbol{z}^{\text{cln}}, \boldsymbol{z}^{\text{adv}}), \tag{5}$$

where $\lambda \in (0, 1)$ is a hyperparameter to control the penalty degree of the two terms, and $\boldsymbol{z}^{\text{cln}}$ and $\boldsymbol{z}^{\text{adv}}$ denote the global representations, *i.e.*, the global average pooling on the encoded representations, for the clean image and the adversarial example, respectively.

The SNN loss is a variant of contrastive loss, allowing for the inclusion of multiple positive pairs. We regard members belonging to the same determined class (*e.g.*, two clean images) as positive pairs, while members belonging to different determined classes (*e.g.*, a clean image and an adversarial example) as negative pairs. We consider a mini-batch of $2B$ samples, with one half being clean images, *i.e.*, $\{(\boldsymbol{x}_i,\ y_i^{\text{det}}{=}1)\}_{i=1}^B$, and the other half of adversarial examples, *i.e.*, $\{(\boldsymbol{x}_i^{\text{adv}},\ y_i^{\text{det}}{=}{-}1)\}_{i=B+1}^{2B}$, the SNN loss at temperature $\tau$ is defined below:

$$\mathcal{L}_{\text{snn}} = -\frac{1}{2B} \sum_{i=1}^{2B} \log \frac{\sum_{i \neq j, y_i^{\text{det}} = y_j^{\text{det}}, j=1,\ldots,2B} \exp(-\text{sim}(\boldsymbol{z}_i, \boldsymbol{z}_j)/\tau)}{\sum_{i \neq k, j=1,\ldots,2B} \exp(-\text{sim}(\boldsymbol{z}_i, \boldsymbol{z}_k)/\tau)}, \tag{6}$$

where $\boldsymbol{z}_i$ is the visual representations for the input $\boldsymbol{x}_i$ and the similarity metric $\text{sim}(\cdot, \cdot)$ is measured by the cosine distance. The SNN loss enforces each point to be closer to its positive pairs than to its negative pairs. In other words, the SNN loss penalizes the similarity level between clean images and adversarial examples, making adversarial examples more discernible by our detector.

### 3.3 CLASSIFICATION

Inspired by self-supervised pre-training for vision tasks Chen et al. (2020); Bao et al. (2022); He et al. (2022), we separate our adversarial training into two stages, *i.e.*, pre-training and fine-tuning, for learning high-quality visual representations and fine-tuning a robust classifier, respectively.

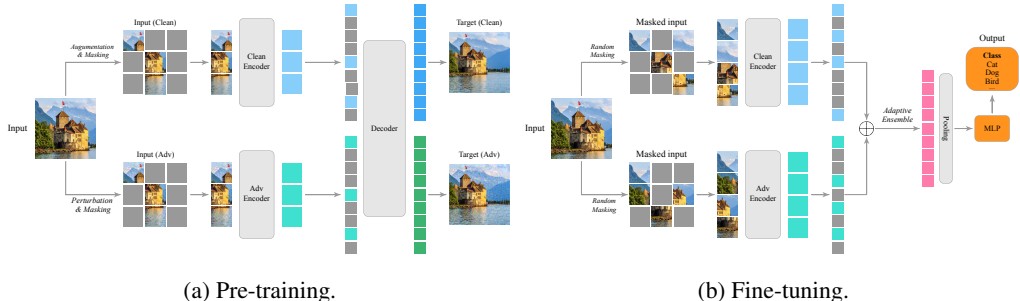

(a) Pre-training.                                          (b) Fine-tuning.

Figure 3: Our classifier architecture: (a) pre-training and (b) fine-tuning.

**Pre-training.** Our architecture for the pre-training is inspired by MAE He et al. (2022). Different from MAE, however, we utilize two encoders, denoted as the clean encoder and the adversarial encoder, for learning visual representations from clean images and adversarial examples, respectively. The decoder aims to reconstruct the original inputs from the visual representations encoded by the two encoders. Figure 3a shows the architecture of our pre-training. Given an input image $x \in \mathbb{R}^d$, let $x^{\mathrm{cln}}$ and $x^{\mathrm{adv}}$ denote its clean and adversarial variants, respectively, with the clean variant obtained by augmenting the original input. Regarding the clean variant $x^{\mathrm{cln}}$, we randomly mask out a large proportion of image patches (*e.g.*, 75%) and then feed the subset of visible patches into the clean encoder. The masked tokens are inserted into corresponding positions after the encoder. Finally, the decoder reconstructs the clean variant $\bar{x}^{\mathrm{cln}}$ from the full set of image patches, including encoded visible patches and masked tokens. The reconstruction of the adversarial variant $x^{\mathrm{adv}}$ follows a similar procedure, except that its visible patches are encoded by the adversarial encoder. Notably, the position of masked image patches in the adversarial variant $x^{\mathrm{adv}}$ is the same as that in the clean variant $x^{\mathrm{cln}}$ in order to minimize their visual representation difference during pre-training.

Denote $\bar{z}^{\mathrm{cln}}$ and $\bar{z}^{\mathrm{adv}}$ respectively to be the global representations of the clean and adversarial variants, *i.e.*, the global average pooling on the input sequence for the decoder. Different from MAE, our design utilizes a new loss function to learn visual representations by simultaneously minimizing the reconstruction error and the visual representation difference, *i.e.*,

$$\mathcal{L}_{\mathrm{enc}} = (1 - \Omega) \cdot \mathcal{L}_{\mathrm{rec}}(x, \bar{x}) + \Omega \cdot \mathcal{L}_{\mathrm{cl}}(\bar{z}^{\mathrm{cln}}, \bar{z}^{\mathrm{adv}}), \tag{7}$$

where $\Omega \in (0, 1)$ is a hyperparameter and $\bar{x}$ is the reconstructed image. $\mathcal{L}_{\mathrm{rec}}$ and $\mathcal{L}_{\mathrm{cl}}$ denote the reconstruction loss and the contrastive loss, respectively. Given a set of $B$ input images, we first generate their adversarial variants, arriving at a mini-batch of $2B$ samples, consisting of $B$ clean variants $\{x_i^{\mathrm{cln}}\}_{i=1}^{B}$ and $B$ adversarial variants $\{x_i^{\mathrm{adv}}\}_{i=B+1}^{2B}$. We consider the form of contrastive loss in SimCLR Chen et al. (2020), and define our contrastive loss at temperature $\tau$ as follows:

$$\ell(i, j) = -\log \frac{\exp(\mathrm{sim}(\bar{z}_i, \bar{z}_j)/\tau)}{\sum_{i \neq k, k=1, \ldots, 2B} \exp(\mathrm{sim}(\bar{z}_i, \bar{z}_k)/\tau)},$$
$$\mathcal{L}_{\mathrm{cl}} = \frac{1}{2B} \sum_{k=1}^{B} [\ell(k, k+B) + \ell(k+B, k)], \tag{8}$$

where $\bar{z}_i$ denotes visual representations for $x_i^{\mathrm{cln}}$ (or $x_i^{\mathrm{adv}}$) and the similarity level $\mathrm{sim}(\cdot, \cdot)$ is measured by the cosine distance. In particular, we regard the clean and adversarial variants from the same input as the positive pairs, while the rest in the same batch are negative pairs. Therefore, our contrastive loss decreases when visual representations for the clean and the adversarial variants of the same input become more similar.

**Fine-tuning.** Figure 3b depicts our architecture during fine-tuning, which only keeps two pre-trained decoders with *frozen weights* for producing visual representations during adversarial training, and the decoders are dropped after pre-training. Different from MAE, which encodes the full set of image patches during fine-tuning, our approach randomly masks out a relatively small proportion of image patches (*e.g.*, 45%), aiming to eliminate the potential adversarial effect if the input is an adversarial example.

Given an input image $(x, y^{\mathrm{cls}})$, where $x \in \mathbb{R}^d$ is either a clean image or an adversarial example with the label $y^{\mathrm{cls}} \in [C]$, we randomly mask the input image twice, arriving at two different masked

inputs. Two subsets of visible patches from the two masked inputs are fed into the clean and the adversarial encoders, respectively. The masked tokens are introduced onto their corresponding positions after the decoder, obtaining two full sets of visual representations, *i.e.*, $\hat{z}^{\text{cln}}$ and $\hat{z}^{\text{adv}}$ which are partially encoded by the clean and the adversarial encoders, respectively. We then perform the global average pooling on the *adaptive ensemble* of $\hat{z}^{\text{cln}}$ and $\hat{z}^{\text{adv}}$, with the result fed into an MLP for classification.

**Adaptive Ensemble.** Although randomly masking an input image can eliminate the potential adversarial effect, this way inevitably hurts standard accuracy during fine-tuning. In this paper, we propose *adaptive ensemble* to tackle this issue. That is, the global representation for an input image is derived from the sum of $\hat{z}^{\text{cln}}$ and $\hat{z}^{\text{adv}}$ with an adaptive factor $p \in [0, 1]$, where $\hat{z}^{\text{cln}}$ and $\hat{z}^{\text{adv}}$ are visual representations encoded by the clean and the adversarial encoders, respectively, and $p$ is the probability of the input image being a clean image estimated by our detector.

Let $A$ be a full set of image patches and $V$ be a subset of $A$, including visible patches only. $\mathbb{1}_V(\cdot)$ is the indicator function for evaluating whether an image patch is visible. Hence, for every image patch of $A$, we have,

$$\mathbb{1}_V(i) = \begin{cases} 1, & \text{if the } i\text{-th patch is visible} \\ 0, & \text{otherwise} \end{cases} \quad, \text{ for } i = 1, 2, \ldots, M, \tag{9}$$

where $M$ is the number of image patches, *i.e.*, $|A|$. For notational convenience, we let $\mathbb{1}_V^{\text{cln}}$ indicate the visible patches fed into the clean encoder. Likewise, $\mathbb{1}_V^{\text{adv}}$ indicates the visible patches fed into the adversarial encoder. Let $\hat{z}_i$ be the visual representation of the $i$-th image patch, with $i \in \{1, 2, \ldots, M\}$. Then, our adaptive ensemble is defined by:

$$\hat{z}_i = \frac{p \cdot \mathbb{1}_V^{\text{cln}}(i) \cdot \hat{z}_i^{\text{cln}} + (1 - p) \cdot \mathbb{1}_V^{\text{adv}}(i) \cdot \hat{z}_i^{\text{adv}}}{\max\left(p \cdot \mathbb{1}_V^{\text{cln}}(i) + (1 - p) \cdot \mathbb{1}_V^{\text{adv}}(i), \epsilon\right)}, \tag{10}$$

where the denominator serves to normalize the adaptive ensemble of $\hat{z}_i^{\text{cln}}$ and $\hat{z}_i^{\text{adv}}$, and $\epsilon$ is a small value to avoid divison by zero (*i.e.*, $\epsilon = 1e - 12$ in this paper). The intuition underlying Eq. (10) is that if our detector has a high confidence that the input is a clean image (*i.e.*, $p$ is large), the global representation $\hat{z}_i$ will be mostly encoded by the clean encoder. Otherwise, $\hat{z}_i$ will be mainly encoded by the adversarial encoder. In addition, as our pre-training encourages the similarity level of the clean and the adversarial variants from a given input (see Eq. (7) and Eq. (8)), and two different masked inputs exist upon fine-tuning, the invisible image patches in one masked input can be glimpsed from the other masked input.

## 4 EXPERIMENTS AND RESULTS

### 4.1 EXPERIMENTAL SETUP

**Datasets.** We conduct experiments on three widely-used benchmarks. (i) **CIFAR-10** Krizhevsky et al. (2009): $60,000$ 32x32 RGB images of 10 classes. (ii) **CIFAR-100** Krizhevsky et al. (2009): $60,000$ 32x32 RGB examples in 100 categories. (iii) **Tiny-ImageNet** Deng et al. (2009): $120,000$ 64x64 RGB images of 200 classes.

**Compared Methods.** We compare our approach with four detection approaches, *i.e.*, **Odds** Roth et al. (2019), **NIC** Ma & Liu (2019), **GAT** Yin et al. (2020), and **JTLA** Raghuram et al. (2021). Meanwhile, to exhibit how our approach boosts adversarial training (AT), we compare our method with six AT counterparts: **PGD-AT** Madry et al. (2018), **TRADES** Zhang et al. (2019), **FAT** Wong et al. (2020), **EAT** Tramèr et al. (2018), **Sub-AT** Li et al. (2022), and **LAS-AWP** Jia et al. (2022). Hyperparameters for the baselines, if not specified, are set as reported in their original literature.

**Evaluation.** We consider both dense attacks and sparse attacks for evaluating detection accuracy and adversarial robustness. In particular, we utilize (i) four dense attacks, *i.e.*, **FGSM** Goodfellow et al. (2015), **PGD** Madry et al. (2018), **DeepFool** Moosavi-Dezfooli et al. (2016), and **AutoAttack** Croce & Hein (2020), and (ii) three sparse attacks, *i.e.*, **C&W** $L_0$ Carlini & Wagner (2017), **SparseFool** Modas et al. (2019), and **FMN** Pintor et al. (2021). For notational convenience, we let PGD-20/50 denote the PGD attack with 20 or 50 steps.

**Parameter Settings.** Inspired by the recent success of Vision Transformer (ViT) Dosovitskiy et al. (2021), we use the ViT as the backbone network for our detector and classifier, with their ViT architectures respectively following DeiT Touvron et al. (2021) and MAE He et al. (2022). But we prune our model size as small as possible in order to conduct a fair comparison with baselines. Due to the page limit, we defer the details of the model size and hyperparameters to Section A.1 in Appendix.

## 4.2 OVERALL PERFORMANCE

Table 1: Overall comparisons on CIFAR-10, with the best results shown in bold

| Method | Standard Accuracy | Dense Attack | | | | Sparse Attack | | |
|---|---|---|---|---|---|---|---|---|
| | | FGSM | PGD-20 | DeepFool | AutoAttack | C&W $L_0$ | SparseFool | FMN |
| PGD-AT | 82.3 | 48.4 | 45.6 | 46.2 | 41.2 | 16.5 | 10.3 | 12.5 |
| TRADES | 84.7 | 52.5 | 45.5 | 46.2 | 42.1 | 14.1 | 9.9 | 12.4 |
| FAT | 84.9 | 51.9 | 45.9 | 48.6 | 40.2 | 10.4 | 13.3 | 12.7 |
| EAT | 83.5 | 52.8 | 50.1 | 47.9 | 47.1 | 16.4 | 18.9 | 16.9 |
| Sub-AT | 80.5 | 52.3 | 51.0 | 49.6 | 48.1 | 19.1 | 16.2 | 11.3 |
| LAS-AWP | 85.6 | 57.1 | 56.3 | 53.9 | **51.3** | 20.5 | 13.8 | 16.1 |
| **Ours** | **91.2** | **57.5** | **56.6** | **54.6** | 49.4 | **54.8** | **52.5** | **51.4** |

**Comparisons to Baselines.** We first conduct extensive experiments on CIFAR-10 and compare our approach to state-of-the-art adversarial training (AT) counterparts listed in Section 4.1 in terms of standard accuracy, robustness against dense attacks, and robustness against sparse attacks. Table 1 lists comparative results. It is observed that our approach achieves the best performance under all three scenarios. In particular, our approach achieves the standard accuracy of $91.2\%$, outperforming the best state-of-the-art (*i.e.*, LAS-AWP) by $5.6\%$. This is contributed by employing two encoders to extract visual representations respectively from clean images and adversarial examples, able to significantly mitigate the adverse effect of adversarial training on standard accuracy. Besides, in terms of robustness against dense attacks, our approach outperforms its all competitors under every scenario except LAS-AWP on robustness against AutoAttack. This is because LAS-AWP employs an automatic attack strategy to produce the threat model, similar to the attack strategy of AutoAttack. It should be noted that our approach significantly outperforms all compared counterparts under all sparse attacks, unlike prior ATs that tend to suffer from robustness overfitting upon dense attacks. Take LAS-AWP as an example, its robustness degrades from $57.1\%$ under dense attack (*i.e.*, FGSM attack) to $20.5\%$ under sparse attack (*i.e.*, C&W $L_0$ attack). By contrast, our approach exhibits only $2.7\%$ robustness degradation under the same case, *i.e.*, $57.5\%$ under FGSM attack versus $54.8\%$ under C&W $L_0$ attack. This evidences that our proposed masked adaptive ensemble can boost adversarial training by generalizing robustness improvement to defend against sparse attacks.

More experiments for overall comparisons on CIFAR-100 and Tiny-ImageNet are also conducted, with the results deferred to Section A.3 in Appendix to conserve space.

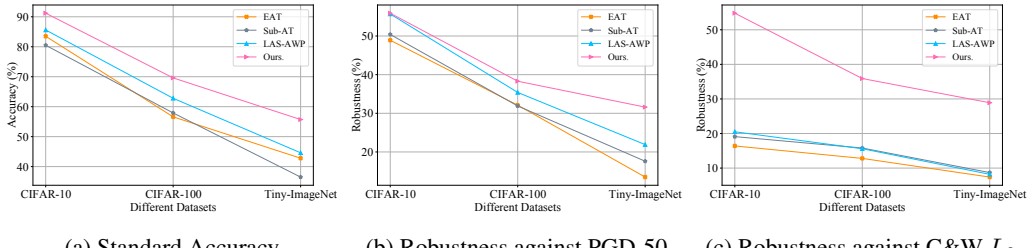

(a) Standard Accuracy      (b) Robustness against PGD-50      (c) Robustness against C&W $L_0$

Figure 4: Illustration of the performance stability under different scales of datasets and different types of attacks.

**Performance Stability.** We next conduct experiments on CIFAR-10, CIFAR-100, and Tiny-ImageNet to evaluate the performance stability under different scales of datasets and different types of attacks. We compare our approach with three baselines, *i.e.*, EAT, Sub-AT, and LAS-AWP. Figures 4a, 4b, and 4c illustrate the comparative results of standard accuracy, robustness against dense attack (*i.e.*, PGD-50), and robustness against sparse attack (*i.e.*, C&W $L_0$), respectively. We have three discoveries. First, as depicted in Figure 4a, our approach suffers from the least standard accuracy degradation of $35.5\%$ (with standard accuracy ranging from $91.2\%$ on CIFAR-10 to $55.7\%$ on

Tiny ImageNet), outperforming EAT (*i.e.*, 39.7% degradation), Sub-AT (*i.e.*, 44.0% degradation), and LAS-AWP (*i.e.*, 41.0% degradation). Second, under the strong dense attack (*i.e.*, PGD-50), our approach and LAS-AWP achieve similar robustness on CIFAR-10, but our method outperforms LAS-AWP by 2.9% on CIFAR-100 and by 9.7% on Tiny-ImageNet, as illustrated in Figure 4b. Third, as shown in Figure 4c, all competitors suffer from poor robustness ($\leq 20.5\%$) under the C&W $L_0$ attack, where our approach still exhibits decent performance, with its robustness equaling 54.8%, 36.8%, and 28.9% on CIFAR-10, CIFAR-100, and Tiny-ImageNet, respectively. These results validate that our approach is more stable when upscaling to large datasets and when defending against different types of attacks.

### 4.3 EVALUATING OUR DETECTOR

Table 2: Comparisons of detection accuracy on CIFAR-10 under dense and sparse attacks

| Method | Dense Attack | | | | Sparse Attack | | |
|--------|------|--------|----------|------------|---------|------------|------|
| | FGSM | PGD-20 | DeepFool | AutoAttack | C&W $L_0$ | SparseFool | FMN |
| Odds | 96.9 | 93.4 | 90.1 | 90.6 | 72.5 | 69.2 | 68.5 |
| NIC | 96.8 | 97.2 | 91.9 | 92.1 | 74.3 | 76.5 | 72.8 |
| GAT | 95.4 | 92.6 | 92.3 | 92.4 | 73.4 | 79.6 | 68.7 |
| JTLA | 97.5 | 95.1 | 93.1 | 93.5 | 77.7 | 74.9 | 69.5 |
| **Ours** | **99.6** | **99.1** | **97.9** | **98.4** | **96.4** | **98.5** | **97.1** |

In this section, we conduct experiments on CIFAR-10, and compare our detector with four detection baselines, *i.e.*, Odds, NIC, GAT, and JTLA. Table 2 lists the detection accuracy values under both dense and sparse attacks. We observed that our detector always achieves superior detection performance under all scenarios, *i.e.*, more than 97.1%, outperforming all competitors. Specifically, our approach achieves the best detection accuracy of 99.6% under the dense attack (*i.e.*, FGSM) and of 98.5% under the sparse attack (*i.e.*, SparseFool). Besides, state-of-the-art counterparts are ineffective to sniff sparse attack-generated adversarial examples, with much inferior detection accuracy. For example, their best detection accuracy under sparse attacks is only 77.7%, *i.e.*, JTLA against C&W $L_0$. In sharp contrast, our approach under C&W $L_0$ yields the detection accuracy of 96.4%, drastically outperforming the best counterpart of JTLA by 18.7%. The statistical evidence exhibits that our two new designs for the detector, *i.e.*, our propose loss (*i.e.*, Eq.(5)) and the Guided Backpropagation-based input variant, are effective for exposing adversarial perturbation, boosting the detector to far better defend against sparse attacks.

Due to the page limit, we defer ablation studies on our classifier (including the impacts of different masking ratios and of our adaptive ensemble) and on our detector (including the effects of Guided Backpropagation and of the SNN loss) to Section A.4 and Section A.5 in the Appendix, respectively.

## 5 CONCLUSION

This article has proposed a novel framework, including a detector and a classifier, to defend against adversarial attacks. With our newly developed adaptive ensemble to bridge the detector and the classifier, our approach can boost adversarial training to defend against both dense and sparse attacks, and can also achieve a better trade-off between standard accuracy and robustness. Our key idea includes applying the Guided Backpropagation to expose adversarial perturbations for better detection and employing two decoders to extract visual representations respectively for clean images and adversarial examples so as to reduce the negative effect of adversarial training on standard accuracy. Meanwhile, our adaptive ensemble allows us to eliminate potential adversarial effects when encountering adversarial examples by masking out a random subset of image patches across input data. Extensive experiments have been conducted for evaluation, with results demonstrating that our solutions significantly outperform their state-of-the-art counterparts in terms of detection accuracy, standard accuracy, and robustness.

Since our approach lies in one-step adversarial training, which typically suffers from "catastrophic overfitting" Wong et al. (2020), where improved robustness may suddenly drop to 0% under the strong PGD attack. We defer the extension of our approach for addressing this possible "catastrophic overfitting" to our future work.

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

## A    APPENDIX

### A.1    MODEL SIZE AND HYPERPARAMETER DETAILS

Table 3: Model Details used in our design

| Model | | Layer | Hidden Size | Head | MLP Size | Params |
|---|---|---|---|---|---|---|
| Detector | DeiT-Tiny | 12 | 192 | 3 | 768 | 5.4M |
| Classifier | Encoder | 12 | 384 | 6 | 1536 | 21.4M |
| | Decoder | 4 | 192 | 3 | 768 | 1.8M |

**Model Size.** To conduct a fair comparison with existing studies, we develop our detector and classifier as small as possible, with Table 3 listing the model size details. Our architecture includes a detector and a classifier with two encoders and one decoder, with $50M$ parameters in total. This is similar to the model size of Wide ResNet Zagoruyko & Komodakis (2016) employed by most previous studies. For example, the Wide ResNet adopted by TRADES Zhang et al. (2019) contains $48.3M$ parameters.

**Hyperparameters.** For all our models, if not specified, we use AdamW Loshchilov & Hutter (2019) with $\beta_1$=0.9, $\beta_2$=0.999, the weight decay of $0.05$, and a batch size of $1024$. We follow the setting in Goyal et al. (2017) to train our detector for $100$ epochs, with the base learning rate of $1e-3$, the linear warmup epochs of $5$, and the cosine decay schedule Loshchilov & Hutter (2017). For our classifier, we pre-train it for $400$ epochs, with the base learning rate of $1e-4$, the linear warmup of $40$ epochs, and a masking ratio of $75\%$. After pre-training, we drop the decoder and freeze the weights on the two encoders. Then, we finetune the classifier for $100$ epochs, with the base learning rate of $1e-3$, the linear warmup of $5$, and the cosine decay schedule, and a masking ratio of $45\%$. The patch size is set to 2 (or 4) for CIFAR-10/CIFAR-100 (or Tiny-ImageNet). We grid-search $\lambda$ in Eq.(5) and $\Omega$ in Eq.(7) and empirically set $\lambda$ to $0.15$ and $\Omega$ to $0.35$ for all datasets.

### A.2    ONE-STEP LEAST-LIKELY ADVERSARIAL TRAINING

This section supports the main paper by presenting the technical detail of one-step least-likely adversarial training.

Adversarial training (AT) improves the model's robustness against adversarial attacks by intentionally feeding adversarial examples into the training set. Given a model $f$ with parameters $\boldsymbol{\theta}$, a dataset with $N$ samples, *i.e.*, $\mathbb{X} = \{(\boldsymbol{x}_i, y_i) \mid i \in \{1, 2, \ldots, N\}\}$, the cross-entropy loss function $\mathcal{L}$, and a threat model $\boldsymbol{\Delta}$, AT aims to solve the following inner-maximization problem and outer-minimization problem:

$$\min_{\boldsymbol{\theta}} \sum_i^N \max_{\boldsymbol{\delta} \in \boldsymbol{\Delta}} \mathcal{L}(f_{\boldsymbol{\theta}}(\boldsymbol{x}_i + \boldsymbol{\delta}),\ y_i), \tag{11}$$

where the inner problem aims to find the worst-case training data for the given model, and the outer problem aims to improve the model's performance on such data. Recently, one-step Fast Adversarial

Table 4: Overall comparisons on CIFAR-100, with the best results shown in bold

| Method | Standard Accuracy | Dense Attack | | | | Sparse Attack | | |
|---|---|---|---|---|---|---|---|---|
| | | FGSM | PGD-20 | DeepFool | AutoAttack | C&W $L_0$ | SparseFool | FMN |
| PGD-AT | 54.6 | 35.3 | 34.7 | 31.9 | 27.5 | 12.3 | 7.4 | 11.3 |
| TRADES | 56.8 | 35.6 | 32.2 | 28.9 | 27.8 | 11.5 | 9.3 | 10.6 |
| FAT | 53.1 | 34.3 | 34.3 | 31.8 | 28.7 | 9.7 | 11.2 | 8.4 |
| EAT | 56.6 | 37.5 | 32.1 | 30.1 | 28.9 | 12.8 | 11.5 | 7.8 |
| Sub-AT | 57.9 | 36.9 | 31.9 | 35.7 | 30.6 | 15.8 | 12.7 | 10.6 |
| LAS-AWP | 62.8 | 38.1 | 35.4 | 36.3 | 29.7 | 15.6 | 11.5 | 11.3 |
| **Ours.** | **69.6** | **40.4** | **38.3** | **38.6** | **37.5** | **36.8** | **35.9** | **34.3** |

Table 5: Overall comparisons on Tiny-ImageNet, with the best results shown in bold

| Method | Standard Accuracy | Dense Attack | | | | Sparse Attack | | |
|---|---|---|---|---|---|---|---|---|
| | | FGSM | PGD-20 | DeepFool | AutoAttack | C&W $L_0$ | SparseFool | FMN |
| PGD-AT | 41.7 | 17.5 | 15.3 | 14.9 | 10.2 | 6.9 | 9.5 | 8.4 |
| TRADES | 36.6 | 18.1 | 15.9 | 13.8 | 12.7 | 7.7 | 8.3 | 6.6 |
| FAT | 42.9 | 19.1 | 15.6 | 14.1 | 12.5 | 7.8 | 6.2 | 5.4 |
| EAT | 43.8 | 18.8 | 13.8 | 12.3 | 11.7 | 7.4 | 4.6 | 6.2 |
| Sub-AT | 36.5 | 20.3 | 18.2 | 13.2 | 11.1 | 8.7 | 9.3 | 6.2 |
| LAS-AWP | 44.6 | 26.5 | 22.5 | 18.3 | 17.5 | 8.2 | 4.7 | 7.2 |
| **Ours.** | **55.7** | **34.2** | **32.2** | **31.7** | **29.6** | **28.9** | **26.8** | **26.1** |

Training (FAT) Wong et al. (2020) is popular due to its computational efficiency. FAT sets the threat model under a small and $l_\infty$ constraint $\epsilon$, *i.e.*, $\Delta = \{\delta : \|\delta\|_\infty \leq \epsilon\}$, by performing Fast Gradient Sign Method (FGSM) Goodfellow et al. (2015) with the random initialization, *i.e.*,

$$\begin{aligned}
\delta &= \text{Uniform}(-\epsilon, \epsilon) + \epsilon \cdot \text{sign}(\nabla_{\boldsymbol{x}} \, \mathcal{L}(f_{\boldsymbol{\theta}}(\boldsymbol{x}_i), \, y_i)), \\
\delta &= \max(\min(\delta, \epsilon), -\epsilon),
\end{aligned} \tag{12}$$

where *Uniform* denotes the uniform distribution and *sign* is the sign function. Notably, the second row in Eq. (12) serves to project the perturbation $\delta$ back into the $l_\infty$ ball around the data $\boldsymbol{x}_i$.

To find the worst-case adversarial examples, we extend FAT by performing the least-likely targeted attacks, inspired by prior studies Kurakin et al. (2017); Tramèr et al. (2018). That is, given an input $\boldsymbol{x}_i$, we perform targeted FGSM by setting the targeted label as its least-likely class, *i.e.*, $y_i^{ll} = \arg\min f_{\boldsymbol{\theta}}(\boldsymbol{x}_i)$, arriving at,

$$\begin{aligned}
\delta &= \text{Uniform}(-\epsilon, \epsilon) + \epsilon \cdot \text{sign}(\nabla_{\boldsymbol{x}} \, \mathcal{L}(f_{\boldsymbol{\theta}}(\boldsymbol{x}_i), \, y_i^{ll})), \\
\delta &= \max(\min(\delta, \epsilon), -\epsilon),
\end{aligned} \tag{13}$$

Our one-step least-likely adversarial training is to utilize Eq.(13) to produce the threat model.

### A.3 OVERALL COMPARSIONS ON CIFAR-100 AND TINY-IMAGENET

This section supports the main paper by comparing our approach with adversarial training (AT) counterparts on CIFAR-100 and Tiny-ImageNet.

Table 4 and Table 5 list the experimental results on CIFAR-100 and Tiny-ImageNet, respectively. From Table 4, we discovered that our approach achieves the best performance in terms of standard accuracy, robustness against dense attacks, and robustness against sparse attacks. In particular, our approach achieves the standard accuracy of 69.6%, outperforming the best competitor (*i.e.*, LAS-AWP) by 6.8%. Besides, our approach achieves the best robustness of 40.4% under the dense attack (*i.e.*, FGSM) and of 36.8% under the sparse attack (*i.e.*, C&W $L_0$). Moreover, existing ATs achieve very poor robustness against sparse attacks (less than 15.8%), while our approach, even in the worst case (*i.e.*, under FMN attack), still maintains a decent robustness of 34.3%.

We have three observations from Table 5. First, our approach outperforms all counterparts under all three scenarios on Tiny-ImageNet. In particular, our approach achieves the standard accuracy of 55.7%, outperforming the best competitor (*i.e.*, LAS-AWP) by 11.1%. Besides, compared to LAS-AWP under robustness against dense attacks, our method achieves the performance improvements ranging from 7.7% (under FGSM attack) to 13.4% (under DeepFool attack). Third, our approach still achieves decent robustness under sparse attacks (*e.g.*, 28.9% under C&W $L_0$ attack), while all competitors perform very poor when encountering sparse attacks (*i.e.*, less than 10%).

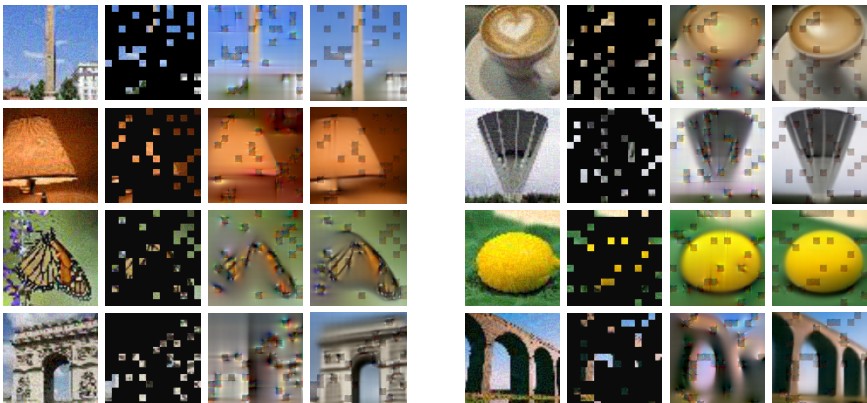

Figure 5: Comparison of the reconstruction quality from masked adversarial examples by employing our approach with/without the contrastive loss, denoted as our approach (w/ CL) and our approach (w/o CL), respectively. From left to right are the original adversarial example, the masked input, reconstruction by our approach (w/o CL), and reconstruction by our approach (w/ CL), respectively.

## A.4 ABLATION STUDIES ON OUR CLASSIFIER

This section supports the main paper by conducting ablation studies on our classifier, including the impact of the contrastive loss on reconstruction quality, as well as the effects of different masking ratios and our proposed adaptive ensemble on the standard accuracy and robustness.

**Pre-training: Contrastive Loss.** We qualitatively and quantitatively exhibit the impact of our proposed loss, *i.e.*, Eq.(7), on learning visual representations. We first present the qualitative evaluations. Specifically, we reconstruct masked adversarial examples and compare reconstruction quality by utilizing our approach with/without the contrastive loss (CL) in SimCLR Chen et al. (2020). Figure 5 illustrates the qualitative results. For images on each row, from left to right, are original adversarial example, the masked input, the image generated by our approach without the CL (*i.e.*, w/o CL), and the image reconstructed by our approach with the CL (*i.e.*, w/ CL). We observed that when using the CL, our approach always achieves a better reconstruction quality; See the 3rd (and 7th) column versus the 4th (and 8th) column. Besides, we discovered that our approach (w/o CL), in some cases, reconstructs adversarial examples with poor quality; See the 3rd and 7th columns in the last row. By contrast, our method (w/ CL) still achieves a high reconstruction quality on these examples; See the 4th and 8th columns in the last row. These empirical results demonstrate that our proposed loss can boost the performance when learning visual representations from adversarial examples.

Following MAE He et al. (2022), we quantitatively evaluate visual representations by using the linear probing accuracy. Specifically, we consider the standard accuracy, the robustness under a dense attack (*i.e.*, PGD-50), and the robustness under a sparse attack (*i.e.*, C&W $L_0$). Table 6a presents the experimental results. We observed that by utilizing the contrastive loss, our approach achieves performance improvement of 2.6%, 11.3%, and 10.7% on the standard accuracy, the robustness against PGD-50, and the robustness against C&W $L_0$, respectively. These empirical results demonstrate the necessity and importance of our proposed loss for learning high-quality visual representations.

**Fine-tuning: Masking Ratio.** In this section, we conduct experiments on CIFAR-10 to explore how different masking ratios affect the performance of our approach during the finetuning. 12 groups of masking ratios are taken into account, ranging from 25% to 80%. Note that in the pre-training, we directly set the masking ratio to 75% by following MAE He et al. (2022); hence, no similar ablation study requires. We consider the trade-off between the standard accuracy and the robustness, including robustness against a dense attack (*i.e.*, PGD-50) and against a sparse attack (*i.e.*, C&W $L_0$).

Figures 6a and 6b illustrate the experimental results. From Figure 6a, we observed that increasing the masking ratio will negatively affect the standard accuracy (*i.e.*, the grey line) in all scenarios. In contrast, when the masking ratio is small (*i.e.*, $\leq 50\%$), a larger masking ratio benefits the robustness against the dense attack (*i.e.*, the blue line). But when the masking ratio is greater than 50%,

Table 6: Ablation studies on our classifier, including (a) pre-training and (b) fine-tuning, as well as on our detector, including (c) Guided Backpropagation

| Method | Standard Accuracy | Robustness | |
|---|---|---|---|
| | | PGD-50 | C&W $L_0$ |
| w/o CL | 75.8 | 37.1 | 36.5 |
| w/ CL | 78.4 | 48.4 | 47.2 |

(a) Pre-training: Contrastive Loss (CL)

| Method | Standard Accuracy | Robustness | |
|---|---|---|---|
| | | PGD-50 | C&W $L_0$ |
| w/o AE | 79.7 | 49.3 | 49.4 |
| w/ AE | 91.2 | 55.9 | 54.8 |

(b) Fine-tuning: Adaptive Ensemble (AE)

| Method | PGD-20 | C&W $L_0$ |
|---|---|---|
| w/o GB | 95.7 | 82.6 |
| w/ GB | 99.1 | 96.4 |

(c) Detection: Guided Backpropagation (GB)

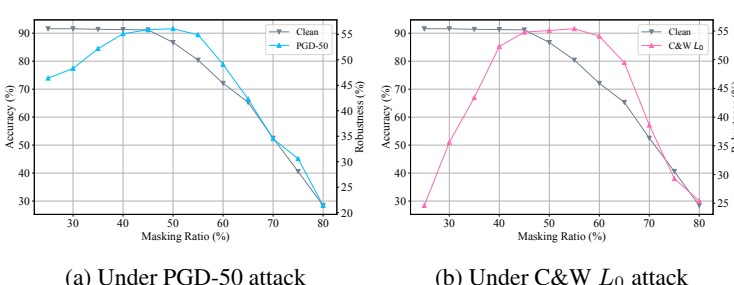

(a) Under PGD-50 attack      (b) Under C&W $L_0$ attack

Figure 6: Illustration of how different masking ratios in the finetuning affect the performance.

increasing the masking ratio hurts the robustness. This is because a small subset of masked patches can eliminate the adversarial effect of adversarial attacks, while a large subset of masked patches would prevent our classifier from accurate classification. Clearly, our approach achieves the best trade-off on the masking ratio of $45\%$ (or $40\%$), with the standard accuracy of $91.2\%$ (or $91.3\%$) and the robustness against PGD-50 of $55.9\%$ (or $55.1\%$).

Similarly, Figure 6b depicts the robustness against the sparse attack (*i.e.*, the pink line) under different masking ratios. We also include the standard accuracy (similar to Figure 6a) for a better illustration of the trade-off. Obviously, when the masking ratio equals $45\%$, our approach achieves the best trade-off, with the standard accuracy of $91.2\%$ and the robustness of $54.8\%$. Based on the above discussion, we can set our masking ratio to $45\%$ to ensure the best trade-off between the standard accuracy and the robustness against both dense and sparse attacks.

**Fine-tuning: Adaptive Ensemble.** Here, we conduct experiments to show the impact of our adaptive ensemble on the standard accuracy and the robustness. Table 6b lists the experimental results with/without our adaptive ensemble. Note that we employ the naive average ensemble when conducting experiments without our adaptive ensemble. From Table 6b, we observed that our adaptive ensemble significantly benefits the standard accuracy, with $11.5\%$ performance improvement. Meanwhile, it boosts adversarial robustness against PGD-50 by $6.6\%$ and against C&W $L_0$ by $5.4\%$. This is because the adaptive factor $p$ in Eq.(10) estimated by our detector can adaptively adjust the proportion of visual representations from clean and adversarial encoders.

## A.5 ABLATION STUDIES ON OUR DETECTOR

This section supports the main paper by conducting ablation studies on our detector. We present the effects of Guided Backpropagation and our proposed loss, *i.e.*, Eq.(5), on the detection accuracy and visual representations for clean images and adversarial examples, respectively.

**Guided Backpropagation on the Detection Accuracy.** In this section, we empirically show how Guided Backpropagation (GB)-based variant affects the detection accuracy under a dense attack (*i.e.*, PGD-20) and a sparse attack (*i.e.*, C&W $L_0$). Table 6c lists the experimental results with/without the GB-based input variant on CIFAR-10. We discovered that removing the GB-based input variant from our design results in a small detection accuracy degradation of $3.4\%$ under the dense attack. In contrast, it incurs a significant detection accuracy drop of $13.8\%$ under the sparse attack. This

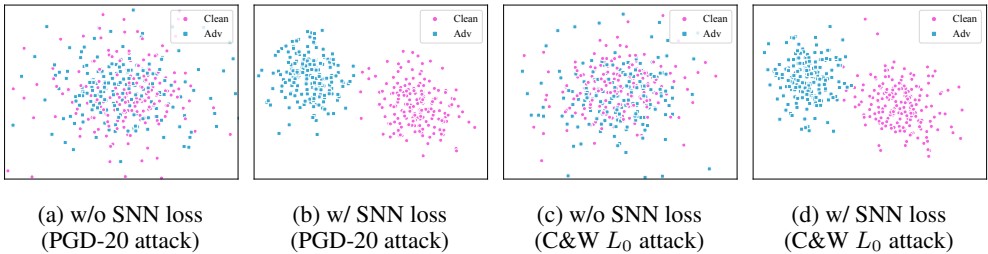

| (a) w/o SNN loss | (b) w/ SNN loss | (c) w/o SNN loss | (d) w/ SNN loss |
| (PGD-20 attack) | (PGD-20 attack) | (C&W $L_0$ attack) | (C&W $L_0$ attack) |

Figure 7: t-SNE visualization on CIFAR-10 by using our detector with/without SNN loss. For each experiment, we perform t-SNE visualization on 200 clean images and 200 adversarial examples generated either by PGD-20 attack, *i.e.*, (a) and (b), or by C&W $L_0$ attack, *i.e.*, (c) and (d).

confirms that Guided Backpropagation can expose adversarial perturbations, making adversarial examples, especially those generated by sparse attacks, easier to be detected.

**SNN Loss on Visual Representations.** This section reveals the effect of our proposed loss, *i.e.*, Eq.(5), on detecting adversarial examples. We consider how our detector with or without the Soft-Nearest Neighbors (SNN) loss affects the resulting representation space. In particular, we employ t-SNE visualization van der Maaten & Hinton (2008) on 200 clean images randomly sampled from CIFAR-10 and 200 adversarial examples generated either by the dense attack (*i.e.*, PGD-20) or by the sparse attack (C&W $L_0$). Figures 7a and 7b depict the results by using PGD-20 attack, while Figures 7c and 7d present the results by employing C&W $L_0$ attack. We observed that without the SNN loss, the representations for clean images and adversarial examples are highly entangled; see Figures 7a and 7c. In sharp contrast, by minimizing the SNN loss, the representations for clean images and adversarial examples are mutually isolated, as shown in Figures 7b and 7d, making adversarial examples detectable.

