# OpenReview forum: "Boosting Adversarial Training with Masked Adaptive Ensemble"
_ICLR.cc/2023/Conference — Submitted to ICLR 2023_

### Official Review · Reviewer_ZoGc · 2022-10-24

**Confidence:** 4
**Correctness:** 3
**Technical Novelty And Significance:** 3
**Empirical Novelty And Significance:** 2
**Recommendation:** 3

**Clarity, Quality, Novelty And Reproducibility:**

Related work:
- Most previous adversarial detection mechanisms can be defeated by a strong, adaptive white-box attacker. This aspect is not addressed in the paper, and the relevant literature ([Carlini&Wagner, 2017], [Athalye et al., 2018]) is not cited.
- Similar classifier-detector combinations have been proposed in the past against adversarial examples (e.g., Magnet [Meng&Chen, 2017]). The paper should cite them in relation to the proposed method.

Soundness:
- The motivation of the paper is not clear: why choose to focus on a mix of dense and sparse attacks? These seem to be a different terminology for $L_{\infty}$ and $L_0$ attacks. It is known that $L_{\infty}$ adversarially trained models are also not robust against $L_2$ adversarial attacks. There is a body of work around defending against the union of perturbation norms. However, to fit into that line of research, the current work would have to also consider at least $L_2$ norm attacks
- Considering that the detector proposed in the paper is based on guided backpropagation and the recent masked autoencoders, the paper should arguably introduce these concepts before building on them.
- The cosine function is not a distance, but a similarity.
- The limitations of the proposed method (e.g., computation cost) are not addressed in the paper. Catastrophic overfitting, which is a known limitation of one-step adversarial training employed by the proposed method is mentioned on the last line of the paper, but left for future work.
- The overall combination of method components (detector and classifier) is not presented in the paper. Sec. 3.3 hints at the fact that the output of the detector is used to fine-tune the classifier model. However, we do not know if the models are trained jointly, separately, in a specific order, etc.

Experiments:
- Important attack parameters are missing from the paper (e.g., step size of PGD). PGD attacks are performed with only 20 steps, resulting most likely in a weak attack that is not proving the strength of the proposed defense.
- No model is evaluated without adversarial training as a baselines. If the attacks are strong enough, the adversarial accuracy of a model trained only with clean samples should be close to zero (sanity check).
- No adaptive white-box attacker is considered.
- FGSM and DeepFool attacks are not very relevant for defense evaluation, as they are weak attacks and outdated. Stronger (e.g., C&W with $L_2$ loss, cited in the paper) and more recent attacks (e.g., ODI [Tashiro et al., 2020]) should be used instead.
- The only strong attack evaluated with $L_{\infty}$ norm is AutoAttack. For this attack, the proposed method does not achieve state-of-the-art performance.
- One would also expect to see more than one model architecture in the evaluation.
- It is unclear if clean and adversarial accuracy of the proposed method are measured considering the agreement between the classifier and the detector in the system (i.e., according to Def. 3.1-3.2).
- It is also unclear if the architecture used to apply the other defenses is the same as used for the proposed classifier (Tab. 1). If not, the comparison seems unfair, as the difference in performance might be coming from the fact that the proposed classifier is a transformer, while the other defenses operate on convolutional neural networks (CNNs).

Clarity:
- The paper is overall well-structured.
- Notations are a bit inconsistent throughout the paper. Some examples:
  * Using the notation $C$ for number of channels and $[C]$ for classes is ambiguous and confusing.
  * Inexact definition of an indicator function.
  * $y_{det}$ in Eq. (5) seems to not have been introduced before.
- The illustration of the detector and the classifier architectures (Fig. 2 and 3) could also include the notations of the elements as used in the text for correspondence.

Reproducibility:
- The code for this paper was not provided.

References:
- [Carlini&Wagner, 2017] Nicholas Carlini, David Wagner. Adversarial Examples Are Not Easily Detected: Bypassing Ten Detection Methods. AISec CCS 2017.
- [Athalye et al., 2018] Anish Athalye, Nicholas Carlini, David Wagner. Obfuscated Gradients Give a False Sense of Security: Circumventing Defenses to Adversarial Examples. ICML 2018.
- [Meng&Chen, 2017] Dongyu Meng, Hao Chen. MagNet: a Two-Pronged Defense against Adversarial Examples. CCS 2017.
- [Tashiro et al., 2020] Yusuke Tashiro, Yang Song, and Stefano Ermon. Diversity can be transferred: Output diversification for white-and black-box attacks. NeurIPS 2020.

**Strength And Weaknesses:**

Strengths:
- Interesting idea to combine the recent masked auto encoders with guided backpropagation for an adversarial example detection framework.
- The combination of ideas in the paper appears to be novel.
- Good ablation study.

Weaknesses:
- Incomplete experimental evaluation, with weak attack baselines and failing to state important attack parameters.
- Limited reproducibility: missing important parameters, code not provided.
- Missing important references related to adversarial examples' detection.

Please see detailed points below.

**Summary Of The Paper:**

This paper proposes a framework for the detection and correct classification of adversarial examples for applications where these cannot be rejected. The focus is the union $L_{\infty}$ and $L_0$ attacks. The entire architecture is based on vision transformers. The detector uses guided backpropagation to render adversarial features more salient. The classifier leverages masked autoencoders, with two encoders. The classifier is trained in two steps, first learning embeddings, then being fine-tuned for the classification task. Experiments are performed on CIFAR-10, CIFAR-100 and Tiny ImageNet.

**Summary Of The Review:**

Interesting and novel idea for a classification and detection framework of adversarial samples. Unfortunately, the experimental evaluation is not strong enough to support the performance claims of the method. Moreover, the method is not set in the context of similar methods literature and their failures.

---

### Official Review · Reviewer_uWvJ · 2022-10-26

**Confidence:** 5
**Correctness:** 2
**Technical Novelty And Significance:** 2
**Empirical Novelty And Significance:** 3
**Recommendation:** 3

**Clarity, Quality, Novelty And Reproducibility:**

Clarity: The whole framework is not clear to me. Also, there is a lack of discussion on several choices of the loss function such as SNN. Although it shows it could improve the detection in appendix, I am not sure why SNN is chosen.

Quality: The overall written is fine.

Novelty: The novelty is mostly existed in the empirical results in my opinion. Also, it is also damaged by the design of their experiments as well.

Reproducibility: The paper has included detailed hyperparameters. However, as there is no clear pipeline, it is hard to judge it is reproducible.

**Strength And Weaknesses:**

Pros:
1. The proposed method achieve a good result in the L_0 attack case without introducing additional training.


Cons:
1. The illustration of proposed framework could be improved. The whole framework is composed by several large components however it is not easy to get the whole picture how to do the training and inference in the end.
2. It is not clear how the detector is trained. Is it trained before the pre-training and fine-tuning? If it is, are the adversarial examples generated from other models?
3. The proposed framework is based on the ViT in both detector and classifier by using the patchify operation. It is not clear whether it could be applied in other CNN models such as ResNet, VGG etc.
4. The improvement on the L-inf attack (dense attack) is limited. The autoattack is the strongest attack among the attack tested and however the proposed method is actually worse than other baselines. Although the improvement over l_0 attack is significant, however, It is also not fair to compare because those models are trained adversarially with L-inf attacks. A fair comparison would those defenses including ensemble of different attacks.
5. It is not clear whether the attack tested have already known the existence of the detector. It would be trivial to defense if the attacker doesn't know the existence of the detector, which could regarded as another way of obfuscated gradient problem.

**Summary Of The Paper:**

The paper proposes to combine the masked patches from both adversarial examples and clean examples to boost the model's robustness against adversarial attacks. Specifically, to differentiate adversarial example from the clean example, it first train a detector based on the heatmap obtained by guided backpropogation. To make the dectector better, It then chooses the cross-entropy loss together with the SNN loss.  Then it proposes to have a two phases classifier to adversarially train the model. It first do a pre-training with a auto-encoder way to reconstruct image and get a better model initialization. It then adds a MLP after performing a global average pooling on the adaptive ensemble between adversarial features and clean features with a coefficient got by the detector. The proposed method is then tested on CIFAR10, CIFAR100 and tiny imagenet dataset in the experiment section.

**Summary Of The Review:**

The paper proposes to use a combination of detector and adversarial training to improve the model's robustness against adversarial attacks. However, the proposed framework is only shown in the ViT models and there exists some problems in the experiment evaluation.

---

### Official Review · Reviewer_W9ju · 2022-10-27

**Confidence:** 4
**Correctness:** 2
**Technical Novelty And Significance:** 2
**Empirical Novelty And Significance:** Not applicable
**Recommendation:** 3

**Clarity, Quality, Novelty And Reproducibility:**

-[clarity] this paper discuss two different tasks with two models, separately.

-[quality] the experimental evaluations are not convincing, and suffers from fairness issues and missing metrics.

-[novelty] the key components used in this paper are mostly not original. However, using them for the tasks being discussed about is ok.

-[reproducibility] Python code is not available. I'm not confident about the reproducibility given the fact that a massive amount of details are put into the supplementary.

**Strength And Weaknesses:**

Strengths

-[insight] The motivation to turn visualization (guided backprop) into a detection method is good

-[effectiveness] the experimental results demonstrate the effectivenss of the proposed method

Weaknesses

-[typesetting, minor] please use \citep and \citet macros accordingly in the manuscript.

-[adaptive attack, major] This paper talks about both defense and detection. However, adaptive attack (On Adaptive Attacks to Adversarial Example Defenses, neurips2020) and attacks against detection methods are not discussed (Adversarial Examples Are Not Easily Detected:
Bypassing Ten Detection Methods, AISec'17). Specifically, the notation of of adversarial robustness (eq.2) is built upon both detection and defense. So as long as one of them is broken, the whole system is compromised. In other words, the worst-case performance of the combination of detection and defense is the weakest one among detection and defense. In that sense, the adaptive attack is more necessary to be discussed.

-[clarity, minor] in section 3.2, it is illustrated that the visualization inspired the detection method. However, perturbations from different attacks may have different characteristics that result in a distinct pattern in the visualization. This section does not explicitly specify whether the observation is applicable to any (or what) kind of adversarial perturbation. In that sense, the foundation of the proposed method is not very well supported.

-[fairness in comparison, major] in section 4.1 "parameter settings", we know that the baseline methods are still using CNNs instead of a transformer like the proposed method. Even if the transformer is trimmed to a lower number of parameters, it is still posing unfair comparisons to previous works. We know that architecture itself is already a big enough variable regarding adversarial robustness. Plus, a transformer is well-suited for patch-wise representations, which may make transformers naturally good against sparse attacks. In that sense, we cannot isolate the contribution of the transformer itself and the proposed defense method in the robustness gain. What if the previous defense methods use the same transformer backbone? That would reveal the true improvement. The current table 1 suffers from unfair comparison and is not convincing.

-[missing evaluation, major] in section 4.3, only "accuracy" is reported for attack detection. However, it is widely known that for a "detection" tasks, reporting merely TPR (true positive rate) or accuracy without reporting any FPR (false positive rate) is not making an experiment convincing. See also (Adversarial Examples Are Not Easily Detected:
Bypassing Ten Detection Methods, AISec'17). A detector that achieves 99% TPR at a high FPR is not necessarily useful.

**Summary Of The Paper:**

This paper presents two methods for two relevant tasks. One is an attack detection method inspired by guided backpropagation. The other is adversarial training using pre-training and adaptive ensemble techniques. The two methods are evaluated on CIFAR-10, CIFAR-100, and Tiny-Imagenet datasets. Experimental results demonstrate the effectiveness of the proposed methods.

**Summary Of The Review:**

This paper presents two straightforward ideas for attack detection and defense. The experiments are not convincing, suffering from fairness issues and missing metrics. I think it is very hard to change the situation during the rebuttal process. So my recommendation is something amid a weak reject and a strong reject.

---

### Official Review · Reviewer_Ty1m · 2022-10-28

**Confidence:** 3
**Correctness:** 2
**Technical Novelty And Significance:** 2
**Empirical Novelty And Significance:** 2
**Recommendation:** 3

**Clarity, Quality, Novelty And Reproducibility:**

There is some novelty, but the clarity and reproducibility need to be improved, as some important details are missing.

**Strength And Weaknesses:**

The proposed approach is interesting. However, I have the following concerns and questions:
1. The authors claim that the adversarial examples are noticeable with Guided Backpropagation visualization in their empirical observation. However, previous work ex: Fooling Neural Network Interpretations via Adversarial Model Manipulation https://arxiv.org/abs/1902.02041 have shown that those visualizations can be manipulated under the white box settings. It is possible for the attackers to control the region that the model focuses on.

2. It is unclear how the authors generate adversarial images to pre-train the encoder without having the classifier ready. The adversarial example should come from attacking an existing model. Are those adversarial images generated from attacking some model else?

3. How are the adversarial images generated during the evaluation? Is the full structure (detector, two branch image encoder) go under white box attack?

4. The authors claim they beat the state of the art under multiple attacks. However, the epsilon budget for those attacks is not specified throughout the article.

**Summary Of The Paper:**

This paper presents a new framework to improve robust accuracy while maintaining standard accuracy.
Since the gradient visualization for the adversarial image looks different from the normal image by the authors’ empirical observation, the authors use the visualization generated by Guided Grad-CAM and the input image as a pair to learn an adversarial example detector.
It produces a probability for how likely an image is to be attacked.
For the classification task, the authors use two distinct branches to classify normal and adversarial images separately, and adaptively ensemble two branches’ results with the probability predicted by the adversarial detector.
They use a pre-training and fine-tuning receipt to ensure the normal image encoder and the adversarial image encoder produce a similar representation for the same image with and without attack.

**Summary Of The Review:**

Interesting approach, but some important details are missing.

---

### Decision · Program_Chairs · 2023-01-20

**Decision:**

Reject

**Justification For Why Not Higher Score:**

 N/A

**Justification For Why Not Lower Score:**

 N/A

**Metareview: Summary, Strengths And Weaknesses:**

This paper presents a new framework to improve robust accuracy. The presentation is nice and the experiment seems solid. But the reviewers find many major issues that can not be easily fixed. All the reviewers agree to reject.